# Insects’ and Farmers’ Responses to Pollinator-Related Habitat Improvement in Small and Large Faba Bean Fields in Morocco

**DOI:** 10.3390/insects16111164

**Published:** 2025-11-14

**Authors:** Youssef Bencharki, Denis Michez, Patrick Lhomme, Sara Reverté Saiz, Oumayma Ihsane, Ahlam Sentil, Insafe El Abdouni, Laila Hamroud, Aden Aw-Hassan, Moulay Chrif Smaili, Pierre Rasmont, Stefanie Christmann

**Affiliations:** 1International Center of Agricultural Research in Dry Area (ICARDA), Station Experimental INRA-Quich, Street Hafiane Cherkaoui, Agdal, Rabat 10090, Morocco; patrick.lhomme@umons.ac.be (P.L.); oumayma.ihsane@ulb.be (O.I.); ahlam.sentil@gmail.com (A.S.); elabdouni.insafe@gmail.com (I.E.A.); l.hamroud@gmail.com (L.H.); s.christmann@fap-research.de (S.C.); 2Laboratory of Zoology, Research Institute for Biosciences, University of Mons, Place du Parc 20, 7000 Mons, Belgium; denis.michez@umons.ac.be (D.M.); sara.revertesaiz@umons.ac.be (S.R.S.); pierre.rasmont@umons.ac.be (P.R.); 3Independent Researcher, Edmonton, AB T6V 1N9, Canada; aawhassan@gmail.com; 4L’Institut National de la Recherche Agronomique (INRA), Regional Center of Agricultural Research of Kenitra, P.O. Box 257, Kenitra 14000, Morocco; csmaili@yahoo.fr

**Keywords:** habitat enhancement plants, farming with alternative pollinators, pollinators, natural enemies, faba bean

## Abstract

This study tested the Farming with Alternative Pollinators (FAP) approach in small and large fields in Morocco, which combines a main crop with marketable habitat enhancement plants to support biodiversity and farmer income. Results showed that both small (300 m^2^) and large (1 ha) FAP fields compared to monoculture fields had more abundance and diversity in terms of pollinators, and natural pest enemies, and higher incomes, though benefits were stronger in smaller fields. Net income increased by 108% in small fields and 36% in large fields. Farmers also recognized the value of marketable habitat enhancement plants as a cost-effective, multi-benefit solution.

## 1. Introduction

The global decline in biodiversity has been documented in various groups of organisms, with insect pollinators being one of the most important groups [1,2]. The decline of insect populations poses a significant threat to ecosystem functioning and, ultimately, human well-being [3,4]. Pollination is a crucial ecosystem service provided mostly by insects, as it ensures plant sexual reproduction and hence the many ecosystem services associated with plants: clean air, clean water, shelter, fuel, food, and medicine [3,5,6]. Insect pollinators are primarily threatened by the reduction in floral resources and nesting sites [7,8,9]. Additionally, habitat fragmentation resulting from urban development and the excessive use of fertilizers and pesticides for crop production has exacerbated the threat [10]. Notably, pesticides have well-known adverse effects on pollinators [11]. Mitigation strategies aiming to halt the decline in pollinator populations have been tested [10]. In Europe, a shift from conventional to organic agricultural practices has been implemented without experiencing yield losses [12]. In low- and middle-income countries, farmers have little awareness of the threats to wild pollinators [13,14]. In Morocco, for example, one reason for this may be the dominance of cereal crops (i.e., wind-pollinated crops) that cover more than 50% of the arable land [15]. Despite this, pollinator-dependent agricultural production has been increasing for decades [10] and holds significant economic value. In Morocco, the estimated value of insect pollination to agricultural production was 1.85 × 10^9^ € in 2019, approximately 1.7% of the Moroccan PIB in 2019 [16].

While shifting from conventional to organic practices could be seen as a challenging transition by farmers, biodiversity-friendly practices can be proposed as a stepping stone [17]. In the European Union and the USA, agri-environmental schemes offer incentives to promote ecological restoration and reward farmers for enhancing plant diversity by seeding wildflower strips [18,19]. However, in low- and middle-income countries like Morocco, planting wildflower strips is often met with resistance from farmers due to concerns about the spread of weeds in their fields [13,14]. An alternative approach known as “Farming with Alternative Pollinators” (FAP) aims to enhance crop diversity and support populations of pollinators and natural enemies of pests by using marketable habitat enhancement plants (MHEPs) [7,13,14,17,20]. The FAP approach considers local MHEPs such as oil seeds, spices, vegetables, berries, medicinal and aromatic plants, and forage plants. FAP can enhance farmers’ yields [13,14,20,21,22] without relying on pesticides and does not depend on governmental payments, in contrast to agri-environmental schemes [23]. The impacts on beneficial insects and income have been measured to provide evidence to farmers. This approach has been tested in Uzbekistan, Zimbabwe, and several countries in North Africa and the Near East [13,14,20,21,22,24,25,26]. The costs for FAP and control fields are similar on average [13,14,20,22]. FAP demonstrated to be beneficial for farmers in those countries. A recent study with smallholders in Morocco showed that and which MHEP in particular could substantially reduce the antagonism between scientists concerned with pollinator protection and farmers focusing on production; MHEP had a high adoption rate (68%) after the trials ended [21]. The research showed that farmers decide from economic and agronomic points of view, the authors recommended taking this into account when researchers suggest pollinator-protection approaches [22]. However, questions remain regarding the effects of field size (i.e., the FAP approach has mainly been tested on relatively small fields). Clough et al. [27] highlighted field size as a major driver of farm land biodiversity; smaller fields support more pollinators and natural enemies of pest species [28]. Furthermore, the effectiveness of ecological intensification was demonstrated to be higher in small fields [29].

In the present study, we used faba beans as the main crop to compare the efficiency of the FAP approach between small (0.03 ha) and large (1 ha) fields. There had already been trials with faba beans in small fields in Morocco, it is an important crop in the agro-ecosystems of this country [20,21,26,30]. The average annual production from 2013 to 2023 was 92,473 tons, as reported by the Food and Agriculture Organization (FAO).

This study has four primary objectives:Measure the impacts of the FAP approach on pollinator abundance and diversity by comparing of “FAP fields” vs. “control fields” at two spatial scales (i.e., small and large fields).Evaluate the diversity and abundance of pests and their natural enemies in FAP and control fields using the same approach.Compare net income between FAP and control fields in small and large fields.Assess farmer perceptions of FAP through a questionnaire survey.

Our central hypothesis was that implementing the FAP approach in large fields (i.e., one ha surface) would result in increases in pollinator diversity and abundance, natural enemy diversity and abundance, and farm net income. These impacts are expected to be lower in large fields compared to small fields because beneficial interactions (e.g., pollination, pest control) are more concentrated and efficient over a limited area [29]. We also hypothesized that this approach would have a positive impact on the farmers’ perception of the system, benefiting both small- and large-scale farmers, because it can enhance biodiversity, increase income, and reduce the dependency on chemical inputs, making the farming system more resilient and sustainable.

## 2. Materials and Methods

### 2.1. Study Sites

The study was carried out in two distinct agro-climatic zones of Morocco: the semi-arid Settat region and the sub-humid Kenitra region. The semi-arid Settat region, located in the Settat-Casablanca area (33°00′ N–7°36′ W, elevation 600 m), has temperate, dry summer, hot summer, and limited rainfall (approximately 300–400 mm per year). Large and small fields were established in the same locality. In the sub-humid Kenitra region (the Rabat-Sale-Kenitra region; 34°02′ N 6°50′ W, elevation 50 m, with annual rainfall of 400–450 mm per year), we selected fields in two different areas: one near Maamora forest (25 km from Kenitra, Morocco) characterized by small farms producing vegetables, fig trees, and avocados, and one farm in Sidi Slimane, where the fields are larger and known for fruit and legume production. The semi-arid region is characterized by flower scarcity resulting from a hot and dry climate and the prevalence of 90% cereal monocultures. In contrast, the sub-humid region is known for fruit production, particularly oranges, and legume crops, especially in Sidi Slimane, where our large field experiments were conducted. In both areas, we selected a total of 32 small (300 m^2^) fields in 2018–2019, and we selected 14 large (one ha) fields in 2021.

### 2.2. Experimental Design

A two-year study spanning from 2018 to 2019 was conducted in small experimental fields (i.e., 10 × 30 m) in both semi-arid and sub-humid zones. In the FAP fields, 75% of the total area was planted with faba beans (the main crop), and the outer one-meter perimeter (25%) was planted with MHEP encompassing oilseed rape, spices, vegetables, medicinal plants, and forage plants (Appendix A). The control fields were planted with 100% faba beans. A randomized block design was employed, consisting of five FAP and three control fields within each zone per year. Each field was drip-irrigated, and all fields were located at least one km apart to sample different pollinator communities. The location within the farm was changed each year based on availability and irrigation facilities, but maintaining the same soil type. Both FAP and control fields were weeded periodically. The MHEPs in sub-humid Kenitra region comprised wild lupine (Fabaceae: *Lupinus luteus*), arugula (Brassicaceae: *Eruca vesicaria*), canola (Brassicaceae: *Brassica napus*), alfalfa (Fabaceae: *Medicago sativa*), chia (Lamiaceae: *Salvia hispanica*), cultivated lupine (Fabaceae: *Lupinus albus*), coriander (Apiaceae: *Coriandrum sativum*), celery (Apiaceae: *Apium graveolens*), and grass pea (Fabaceae: *Lathyrus sativus*). Wild lupine, grass pea, alfalfa, canola, coriander, arugula, chia, and flax (Linaceae: *Linum usitatissimum*) were planted in semi-arid Settat region.

In the small fields at both locations (i.e., Settat and Kenitra), the selection of MHEPs was based on landscape characteristics and the farmers’ recommendations [20,21]. However, we made changes to the selection of MHEPs in the second year to address various factors, including seed availability, flowering time, and farmer preferences. In the first year, we used a total of ten MHEPs in the small fields. However, this number was reduced to seven in the second year. This reduction was due to several species of plants not exhibiting the required features for attracting pollinators or natural enemies, such as high floral coverage and/or good plant height [21]. Additionally, some plants had short blooming periods. In the semi-arid region, lupine, grass pea, and alfalfa were replaced with other MHEPs such as coriander, chia, flax, and arugula [26]. In the sub-humid region, we removed alfalfa based on farmer feedback and because it was unsuitable for the region.

All the large FAP faba bean fields (1 ha) were planted in both regions in 2021 using just two MHEPs, canola and coriander. In Settat, eight fields (five FAP and three controls) were planted, and six fields (three FAP and three controls) were planted in Sidi Slimane (Appendix A). MHEPs in large FAP fields were planted in seven parallel 1 m × 100 m strips, 15.5 m apart, covering a total area of 0.07 ha (7% of the field). The control fields were planted with 100% faba beans. In large fields, the selection of MHEPs was based on the benefits they provided to crops from FAP trials with small fields conducted in 2018–2019 [20,21].

### 2.3. Pollinator Sampling

Pollinators were sampled three times during the blooming season of both MHEP and the main crop in both small and large fields using sweep nets and pan traps. Each sampling event was carried out between 10 am and 4 pm under suitable weather conditions for foraging visitors (temperatures above 16 °C, clear skies, and calm winds). For each sampling round, data collection occurred over two consecutive days, with each field sampled once, contingent on weather conditions.

In the small fields, sweep netting was processed as follows: for faba beans (75% of the area of the field), two transects 28 m long × 2 m wide were surveyed for five minutes each, resulting in a total sampling time of ten minutes per field. In the other 25% area of each field (i.e., MHEP in FAP or main crop in control fields), pollinators were collected along an 80 m long, one-meter-wide transect for ten minutes.

We conducted sweep netting sampling in the large fields. Two transects were established within each field separately, measuring 100 m long × 15.5 m wide, one in the center and one along the edge. Main crop transects were sampled for ten minutes in both FAP and control fields separately. We sampled pollinators along a 100 × 1 m transect for ten minutes in 7% of the area of the control fields and 7% of the area of the FAP fields planted with MHEPs (five minutes in coriander and five minutes in canola). Sampling time was equal in both field scales (see Appendix A).

Sampling primarily focused on three groups of pollinators: bees, hoverflies, and wasps. All insects were collected, except for honeybees (*A. mellifera*), *Bombus terrestris*, and *Xylocopa pubescens*, which were counted as they are identifiable in the field. Collected bees were anesthetized and killed using jars containing ethyl acetate.

Additionally, we captured entomofauna from all fields using three colored pan traps (500 mL, with a diameter of 145 mm and depth of 45 mm), which were painted white or with UV-reflecting yellow or blue (Rocol Top, Mons, Belgium). Each pan was filled halfway with odorless soapy water to capture entomofauna that may not have been caught by the sweep nets. Two sets were placed in the small fields (two sets were placed within the faba bean fields), and four sets were placed in the large fields (two in the center and two at the edges). The pan traps were set in each field at 10 am and collected after 30 h at the end of each sampling session, i.e., on the following day at 4 pm.

After all specimens were pinned and labeled, we identified them at the family level. For bees, these were Apidae (*Ammobates*, *Anthophora*, *Apis*, *Bombus*, *Ceratina*, *Eucera*, *Melecta*, *Nomada*, *Thyreus*, and *Xylocopa*), Megachilidae (*Anthidium*, *Chelostoma*, *Hoplitis*, *Megachile*, and *Osmia*), Andrenidae (*Andrena*, and *Panurgus*), Colletidae (*Colletes*, and *Hylaeus*), Halictidae (*Ceylatulitus*, *Dufourea*, *Halictus*, *Lasioglossum*, *Seladonia*, *Sphecodes*, and *Systropha*), and Melittidae (*Melitta*, and *Dasypoda*) [31]. For hoverflies, the families were Syrphidae (*Ceriana*, *Episyrphus*, *Eristalis*, *Eristalinus*, *Eupeodes*, *Myolepta*, *Melanostoma*, *Meliscaeva*, *Platynochaetus*, *Scaeva*, *Sphaerophoria*, and *Syritta*), and for wasps, the families were Chrysididae (*Chrysura*, and *Pseudochrysis*), Crabronidae (*Cerceris*, *Diodontus*, *Ectemnius*, *Lindenius*, *Pseudochrysis*, *Lindenius*, *Liris*, *Oxybelus*, *Philanthus*, and *Tachysphex*), Scoliidae (*Campsomeriella*, *Dasyscolia*, and *Megascolia*), Tiphiidae (*Tiphia*), Vespidae (*Euodynerus*, *Polistes*, *Vespa*, *Vespula*, *Vespida*, and *Eumenes*), Tenthredinidae (*Tenthredo* and *Tabidus*). All insect specimens were then sent to specialists for identification to species level (see the Acknowledgements Section).

### 2.4. Pests and Natural Enemies

Pest and natural enemy sampling were conducted in both the main crop and MHEP areas of “FAP fields” vs. “control fields”. The sampling were performed on three occasions for each field in conjunction with the pollinator sampling. We employed the plant beating method to collect pests and natural enemies. In the FAP fields, ten plants were randomly selected from the main crop across the entire field, as well as ten plants from each MHEP. In the control fields, sampling was limited to the main crop, with thirty plants selected at random from the entire field. To ensure consistency in the sampling, each plant was struck ten times with a stick fitted with a rubber cover at the end. The insects that fell into the trap were collected in a plastic bag. After sorting the samples using binocular dissecting scopes [32], the collected samples were pooled into 2 mL Eppendorf tubes containing 70% ethanol. Afterwards, all specimens were sent to specialists for identification (see the Acknowledgements Section).

### 2.5. Assessing Average Net Income from FAP and Control Fields

The calculation of net income for large fields followed the methods developed for small fields [13,19] see for details. Also for large fields, the net income of the main crop was determined by considering the number of faba bean pods and their weight. The number of pods and the weight of faba beans were measured within ten randomly chosen 1 m^2^ quadrats. Then, we extrapolated the count and weight of faba bean pods for the entire field (0.93 ha). The income generated from the 93% zone of the main crop was computed by multiplying the total weight by the local market price per kilogram. Farmers recorded the total yield weight of MHEP (7% zones) within FAP fields, as well as the corresponding control area (7% zones) (faba beans). The total income for the 7% zone of both treatments was calculated by multiplying the total weight by the prevailing per-kilogram market price. To arrive at the final income, we accounted for the respective investment expenses in seeds and the additional labor required for cultivating the MHEP within the 7% zones of FAP fields. This labor cost was estimated at 200 MAD, equivalent to two person-days per field, specifically for MHEP harvesting. The labor expenses for harvesting the 7% zones in control fields were omitted, considering that they could be efficiently harvested with the main crop and would not need extra labor (see Appendix A).

### 2.6. Farmers’ Perceptions of the FAP Approach

Between 12 February and 17 March 2023, a comprehensive set of standardized face-to-face interviews was conducted, encompassing a diverse group of farmers from Morocco (see questionnaire in Appendix A), all of whom had participated in FAP research by cultivating a FAP field at least once, and in some cases, multiple times, between 2018 and 2021. The survey involved 18 small-scale farmers and 28 larger-scale farmers, comprising 45 male farmers and one female farmer from two distinct agro-ecosystems. The interviews were conducted in Arabic. In terms of educational background, the participants exhibited diverse levels of education: 28% were illiterate, 26% had completed primary education, 30% had finished high school, and 15% received a university education. On average, the farmers owned 6.07 hectares of land, with individual holdings ranging from 0.3 to 20 hectares.

### 2.7. Statistical Analysis

The data analysis was performed using R version 4.4.0 [33]. To assess the attractiveness of different field layouts (FAP and control) to various pollinator groups, we pooled the data of all FAP fields and of all control fields for each scale and employed a Generalized Linear Model (GLM) to compare the abundance of the assessed functional groups of pollinators (short-tongued bees, long-tongued bees, hoverflies, and wasps). A negative binomial was used for the data distribution. The model incorporated the treatments “FAP fields” vs. “control fields” as the primary explanatory variables. After analyzing the abundance of functional pollinator groups in MHEP, we performed a pairwise post hoc comparison using Type-3 ANOVA. The packages used for the analysis were lme4 [34] for fitting the GLM, Emmeans [35] for calculating estimated marginal means, and AER [36] for additional support.

For the following analyses, we used a one-way ANOVA to evaluate whether there was a significant interaction between the independent variables in combination. We tested “Treatment” for its effect on the mean abundance and diversity of natural enemies and pests. QQ plots and the Shapiro–Wilk test were used to check for normality, and homogeneity of variance was assessed by Levene’s test from the car package (version 3.1-2) [37]. For the comparison of abundance and diversity between treatment types, we used the Emmeans test function from the rstatix package (version 0.7.2) [38]. Tukey’s test (PAST 4.13) was used to analyze the collected data of the survey.

## 3. Results

### 3.1. Pollinator Abundance and Diversity in Small and Large Fields

#### 3.1.1. Field Level

In the small fields planting incorporated a mosaic design comprising four to six MHEPs in the 25% zone of FAP fields, with two MHEPs in the 7% area of FAP fields in large fields. The FAP fields provided nectar and pollen for a longer period on average, 75 days in FAP fields and 47.5 days in control fields in small fields. In large fields, this period lasted from 64 days in FAP fields to 58 days in control fields. Beneficial insects, including pollinators, were only attracted to the control fields during the flowering period of the main crop. However, FAP fields were attractive forage sites both before and after the main crop flowering period due to the flowering of the MHEPs before, during, and after the main crop, as indicated by the insect sampling (Table 1). During the blooming periods of the main crop and MHEPs, sweep netting and pan traps recorded a diverse array of flower-visiting insects, resulting in a total of 6502 specimens collected and observed across small and large fields (Table 2 and Table 3).

In the small fields, we collected and observed a total of 3925 specimens from both of “FAP fields” and “control fields”. In the semi-arid region [26], short-tongued bees (52% Andrenidae, 44% Halictidae, 3% Colletidae, and 1% Melittidae), the long-tongued bee group (91% Apidae and 9% Megachilidae); wasps (53% Tiphiidae, 42% Scoliidae, 3% Crabronidae, 1% Vespidae, and 1% Chrysididae), and diptera (100% were in the family Syrphidae). In the sub-humid region, of 2280 specimens collected, long-tongued bees (97% Apidae, 3% Megachilidae); short-tongued bees (51% Andrenidae, 38% Halictidae, 10% Colletidae, and 1% Melittidae); wasps (59% Vespidae, 27% Crabronidae, 12% Scoliidae, 1% Chrysididae, and 1% Tiphiidae), and diptera (100% were in the family Syrphidae). Overall, we found a significant difference between FAP and control fields in pollinator richness (*p* < 0.05). (see Appendix A).

In the large fields, a total of 2577 specimens were captured from of “FAP fields” and “control fields” in both regions. In the semi-arid region, long-tongued bees (99% Apidae and 1% Megachilidae); short-tongued bees (91% Andrenidae and 9% Halictidae), diptera (100% in the family Syrphidae), and wasps (37% Tiphiidae, 28% Tenthredinidae, 24% Scoliidae, 6% Vespidae, and 5% Crabronidae). In the sub-humid region, long-tongued bees (99% Apidae and 1% Megachilidae); short-tongued bees (80% Andrenidae, 14% Halictidae, 6% Colletidae); diptera (100% were Syrphidae), and wasps (82% Tenthredinidae, 9% Scoliidae, 6% Vespidae, and 3% Crabronidae). Overall, we found a significant difference between FAP and control fields in pollinator richness (*p* < 0.05) (see Appendix A).

#### 3.1.2. MHEP Level

In small fields, 53% of the hoverfly specimens were attracted to coriander, 25% to canola, and 22% to arugula. The long-tongued bees demonstrated a clear preference for canola, with 78% of the specimens recorded, followed by arugula with 13% and coriander with 9%. Short-tongued bees were attracted to canola (49%), followed by coriander (30%) and arugula (21%). Similarly, wasps were attracted to coriander (59%), canola (31%), and flax (10%). The statistical analysis revealed a highly significant difference in the species richness of pollinators among the MHEPs (*p* < 0.001) (Figure 1). In large fields, the pollinators captured on canola comprised 50% of the specimens, while coriander also attracted 50% of the specimens. There was no significant difference between MHEPs in terms of species diversity (*p* > 0.05) (Figure 2).

#### 3.1.3. FAP Versus Control

The number of pollinators in small fields was significantly higher in “FAP fields” vs. “control fields” (*p* < 0.05). Moreover, the FAP fields exhibited considerably higher species richness, in comparison to the species observed in the control fields (*p* < 0.01) (Figure 3). In large fields, the abundance of pollinators was significantly higher in “control fields” vs. “FAP fields” (*p* < 0.001). Regarding species richness, there was a significant difference in “FAP fields” vs. “control fields” (*p* < 0.05) (Figure 4).

### 3.2. Pests and Natural Enemy Abundance and Diversity in Small and Large Fields

In small fields, earlier research demonstrated a highly significant difference in the pest abundance and diversity of “FAP fields” vs. “control fields” [13,14,17,20,21,25,39]. However, in large fields, a total of 272 natural enemies and 1202 pest specimens were collected in FAP fields, whereas in control fields, 16 natural enemies and 1520 pest specimens were collected. This finding showed a significantly higher abundance of harmful insect populations in the control fields compared to FAP fields (*p* < 0.05). However, in contrast to the results from small fields [14,17,20], there was no significant difference in beneficial insects of “FAP fields” vs. “control fields”. (*p* > 0.05). Additionally, there were no significant differences in the diversity of pests (*p* > 0.05) or natural enemies (*p* > 0.05) of “FAP fields” vs. “control fields” (Figure 5).

### 3.3. Net Incomes from FAP and Control Fields in Small and Large Fields

In large fields, the net income was significantly higher in FAP compared to control fields in both regions. The average net income was calculated based on the weight of faba bean pods and in FAP fields, the yield of MHEP. We counted the weight of faba bean pods in ten randomly selected 1 × 1 m^2^ quadrants. Based on these results, we calculated the weight of the faba bean pods across each field. The income from the 93% zone of the main crop was calculated by multiplying total weight by the market price per kg in 2021. For MHEP (7% zones) in FAP fields and the equivalent area (7% zones) in control fields, FAP farmers weighed and recorded the yield (seeds) of each MHEP, and control farmers weighed and recorded the total weight of faba bean. Income was calculated by multiplying total weight by the market price per kg in 2021. We deducted respective investment costs for seeds and extra-work required for MHEP cultivation in the 7% zones of FAP fields, estimated to be 200 MAD (two persons per day per field) as labor costs for MHEP harvesting. We did not take into consideration labor costs for harvesting the 7% zones in control fields as they can easily be harvested together with the main crop. We calculated the MHEP income of farmers per hectare for FAP fields in comparison to monoculture fields. Based on the economic assessment calculations, the income from FAP fields in both regions is higher by 36% compared to control fields (see Appendix A). There was a significant difference between FAP and control fields (*t* = 2.24, one-tailed *p* = 0.02 and two-tailed *p* = 0.04) (see Appendix A).

Similar analyses for small fields in earlier research showed income was 108% higher in FAP versus control fields in both regions due to the increase in primary crop production [20] (Table 4).

### 3.4. Farmers’ Perceptions of FAP Practice

We interviewed both small and large-scale farmers who had established the FAP approach and had experience with different crops (Appendix A). Of 28 large-scale farmers, the majority cultivated melons, faba beans, and wheat. The 18 small-scale farmers grew faba beans, zucchini, eggplant, melons, and tomatoes. After trials had ended, among the 46 participating farmers, only five small and three large-scale farmers could not establish FAP without assistance from the scientific team. Farmers attributed different benefits to MHEPs. Among these attributes, “induces higher productivity” emerged as the top priority for farmers. When they were asked to rank five different attributes and label them as high, medium, or low importance, the farmers regarded MHEPs as financially efficient instruments that brought advantages to both the field and the entire farm. Most of the farmers ranked “increase quality of the main crop” as “high”, while some farmers ranked “increase in productivity of the main crop” as “high”. In terms of investment costs in MHEPs, the farmers ranked this as “low”. Additionally, farmers ranked “generating income from MHEP” as “medium”. Furthermore, all farmers ranked “lower pest abundance in MHEP” as “medium” attributes.

In response to the question “Would you be ready to seed weeds instead of MHEP to conserve insects?”, all farmers responded with “no”. In contrast, they agreed to use MHEP corridors in pollinator-independent crops to contribute to pollinator protection and benefit from pest control.

When asked about the lessons learned from FAP-trials (an open-ended question with multiple answers possible; see Figure 6), most of the farmers reported that they had discovered how to increase their income through FAP planting designs. Additionally, some learned about the role of protecting pollinators in their fields; 22% gained insights about crop diversification; some became more aware of pollinators and their importance, and some acquired knowledge about safeguarding plants.

All the farmers interviewed acknowledged that Morocco is prone to drought and faces water scarcity due to climate change. A total of 83% of small-scale and 93% of large-scale farmers expressed their readiness to dedicate parts of their fields every 2 km to cultivate perennial crops such as cacti, fruit trees, olives, and medicinal plants. This would ensure a conducive environment for nesting and the offspring of pollinators [7]. However, 17% of small-scale and 7% of large-scale farmers opposed this idea. Overall, most farmers seemed aware of or were interested in the FAP approach. Their main objectives were to increase income, protect pollinators, and diversify crops, rather than relying on pesticides.

## 4. Discussion

### 4.1. FAP Enhances the Abundance and Diversity of Pollinators in Faba Bean Fields

FAP in small and large fields was much more attractive to pollinators compared to control fields, consistent with the findings of previous studies [13,14,20,25,26,30,39,40]. This positive impact can be explained, at least in part, by the prolonged flowering duration in FAP fields achieved through diverse MHEPs and their staggered phenology [13,14,20,26,30,39,40]. Moreover, the diversity of floral traits, such as nectar accessibility, apparently matched the requirements of different functional groups [41]. Our results are in accord with literature on other FAP trials [7,13,14,17,20,26], also reported that FAP in small fields hosted more diverse and abundant pollinators. Overall, this demonstrates that the FAP approach has a positive contribution to species abundance and richness, in both larger fields as well as in the smaller fields. These findings suggest that the FAP approach may be effective and adaptable in different agricultural landscapes, corroborating the results of FAP-cucumber trials in Uzbekistan [13] and Morocco [20] and trials from earlier work in four highly differing agroecosystems (semi-arid, sub-humid, mountainous, and oasis) in Morocco [20,24]. There were also trials in Zimbabwe [24].

### 4.2. Habitat Enhancement Plants and Different Functional Groups

In both small and large fields, we considered farmers’ recommendations for MHEP selection. Coriander and canola were identified as the most promising options due to their ability to attract pollinators and natural enemies of pests, while also providing a reasonable income, although not the highest. Previous studies [20,21] demonstrated that coriander had the highest beneficial potential and was recommended by pollinator specialists, experts on plant protection, and farmers as well; therefore, in Morocco it is considered the MHEP with highest potential to bridge the gap between biologists concerned with pollinator protection and farmers focusing on income [21]. Coriander is highly effective in attracting pollinating insects [42], while canola is noted for its attractiveness to *Apis mellifera* [43] and solitary bees [44]. MHEPs such as coriander, canola, and arugula are highly attractive to pollinators in both regions in small fields, with the sub-humid region showing a higher level of attraction [21,22,39,40,45,46]. This combination extends the bloom period, as the plants offer various types of nectar and pollen. Their tall growth habit (ranging from 0.5 to 1 m) makes them suitable for attracting a greater abundance and diversity of pollinators to the main crop, faba beans. Faba bean plants are primarily visited by long-tongued bees, particularly *Anthophora fulvitarsis* and *Eucera nigrilabris* [26].

### 4.3. Pest and Natural Enemy Abundance and Diversity

The FAP approach, whether in small and large fields, has gained attraction in recent years as a biological control strategy [13,14,17,20,21,39]. The utilization of natural enemies for pest management is of paramount importance for the agricultural sector as a means of increased agricultural production [47]. Controlling pests through natural enemies is not only ecologically significant but also economically practical [48]. The present study demonstrated that pest control was more effective in small and large FAP fields compared to control fields due to the higher pest population in the former. This positive effect of FAP at both spatial scales is likely attributable to the variety of flowers provided by MHEP and/or the diversity of floral nectar. This finding aligns with previous studies [49,50,51] that also examined diversity in the field. Some studies, such as [52], reported that a diverse array of non-crop habitats can improve ecological control of pests. This is because herbaceous and wooded habitats can increase the populations of natural enemies by up to 71% [52]. However, farmers employing alternative farming methods such as FAP do so because this aligns with their land as their most valuable asset [13,20,21]. Our research indicates that planting diverse MHEPs could be an effective way to enhance natural enemies of pests, thus warranting consideration within the framework of integrated pest management for faba bean crops. MHEPs could also be used as push-pull crops [17,39] to reduce pest damage. To ensure the effectiveness of this approach, it is essential to conduct thorough testing over several years in diverse farming systems. Such studies will provide the necessary data to verify the approach’s efficacy and encourage its adoption. It is important to note that it takes time for natural enemies to become established and to start controlling pests [53]. Furthermore, the effect of biodiversity on agroecosystem functioning may not be immediate and can often be observed in the responses of species to predation, competition, or parasitism [54].

### 4.4. Income Benefits from FAP Fields

In this study, faba bean field income increased by 36% in one-hectare FAP fields compared to monoculture fields within just one year. This is a substantial increase, but much lower than the increase described for small fields in earlier research (112% increases in four highly different agro-ecosystems in Morocco) [14,20,21]. Our findings demonstrate the success of FAP regarding farmers’ incomes across diverse agro-ecosystems in Morocco [20,30], despite variation in bee fauna [55]. Moreover, the greater visitation diversity of wild pollinators resulted in higher yields [13,14,20,21,56]. Previous research has also shown the positive impact of FAP on income from fields with various main crops [20,21,22,25,30,39,40]. For example, after cucumber and sour cherry FAP trials in Uzbekistan and Morocco, other farmers adopted FAP methods [13,14,20,21]. Compared to wildflower strips, MHEPs generated income that provided insurance in case the main crop failed due to flooding or other unexpected events [3,13,14,20,21]. Given that small-scale farmers in low- and middle-income countries typically lack access to subsidies that encourage biodiversity, the FAP approach’s emphasis on generating income from marketable crops is a promising strategy for realizing the dual goals of maintaining biodiversity and economic stability [3,13,14,20,21,22].

### 4.5. Farmers’ Point of View Concerning the FAP Approach

This study provides valuable insights into farmer perceptions of the FAP approach in Morocco and confirms earlier results of studies concerned with smallholder farmers [13,14,20,21]. The findings reveal that Moroccan farmers primarily value the agronomic and economic benefits of FAP rather than its ecological or ethical dimensions. This is consistent with other studies in Uzbekistan [13] and Morocco that have shown that improvements in immediate livelihood often take precedence over environmental concerns [57]. The high satisfaction rates reported by both small-scale (89%) and large-scale (57%) farmers suggest that FAP is generally perceived as beneficial. However, satisfaction does not necessarily guarantee long-term adoption. The primary challenges to farming, especially drought and water scarcity, remain pivotal factors concerning how infrastructure and climatic conditions impact the scalability of agroecological innovations such as FAP. All farmers are aware of the challenges posed by climate and water scarcity. Interestingly, while most of the farmers rejected the idea of sowing wildflowers due to concerns over weed proliferation, they expressed strong support for MHEPs, as they avoid opportunity costs; this result is in line with earlier research [13,14,20,24]. This suggests a preference of Moroccan farmers for interventions perceived as controllable and beneficial to production rather than those primarily serving ecological functions and confirms results with Uzbekistan farmers [13]. The result also emphasizes the importance of carefully framing conservation-related components in a way that resonates with farmers’ priorities [21]. Furthermore, the dual use of MHEPs, both for household consumption and market sale, demonstrates the potential of FAP to contribute to both food security and income generation [14,20,21]. This multifunctionality could be an additional incentive for broader adoption, especially among smallholders. Nevertheless, farmers acknowledged the critical issue of pollinator extinction and recognized the economic value of pollination ecosystem services, as pointed out by [16,58], as well as the losses in yield associated with pollinator decline [59]. According to Anougmar [16], the awareness among Moroccan farmers and consumers concerning the value of pollinators increases the more arid the region is. However, a lack of knowledge remains a constraint in Morocco and other low- and middle-income countries [14]. Increasing the accessibility of information to stakeholders is crucial in facilitating the transition to more sustainable practices. It is noteworthy that FAP farmers have voluntarily banned the use of insecticides [20,21]. These results underscore farmers’ perception of MHEPs as a valuable and cost-efficient tool, providing multiple benefits to their fields. Moreover, the questionnaire results indicated that over 72% of small-scale and 64% of large-scale farmers have reduced their pesticide usage due to the positive impact of FAP on the diversity and abundance of pollinators, as well as pest control [14]. The lower pest abundance in the main crop of FAP small fields (64% reduction) demonstrates that FAP can contribute to reducing pesticide dependency among farmers [17,20,21]. A similar trend was observed in large fields, where FAP fields exhibited a reduction in pest abundance compared to control fields. This may encourage farmers to conduct more FAP trials in large fields. However, it is important to note that natural enemies require time to establish and effectively suppress pest populations [53]. Therefore, minimizing pesticide usage is crucial for long-term success. However, a lack of knowledge about alternative methods may still pose a challenge. FAP emerges as a valuable alternative, since it helps protect pollinators and positively impacts pest control, even for pollinator-independent crops [39]. In the farmers’ responses to the questionnaire, faba bean was the most recommended main crop (31%). This preference is likely influenced by the higher income generated by FAP fields. FAP fields yielded 108% and 36% higher income in small and large fields, respectively, compared to control fields [20] (Table 4 and Table 5).

## 5. Conclusions

This comparative analysis should encourage more large-scale FAP on-farm trials investigating other main crops and in more countries, notably in low- and middle-income countries. FAP appears to be economically self-sustaining also in large fields of pollinator-dependent crops, as demonstrated by the examples of melons [40], faba beans (the present study), and sorry cherry [13]. Farmers’ responses have been predominantly favorable. This is important, as in the end, researchers can only make suggestions, while farmers decide whether pollinator protection is worthwhile [21]. A survey on the capacity of agricultural advisors could also offer valuable insights for future pollinator-protection programs.

## Figures and Tables

**Figure 1 insects-16-01164-f001:**
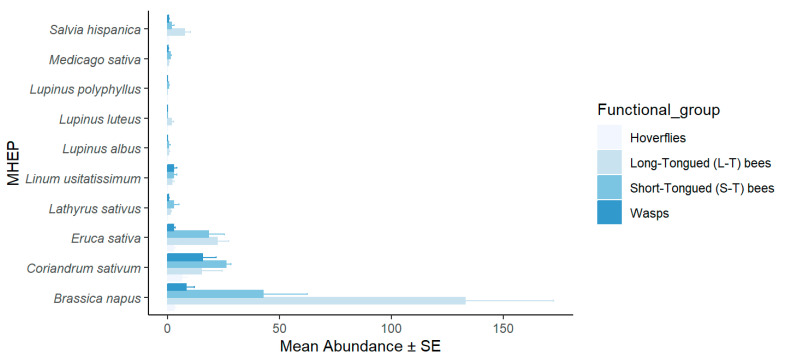
Mean abundance (±SE) of pollinator functional groups visiting each marketable habitat enhancement plant used in small fields in 2018–2019.

**Figure 2 insects-16-01164-f002:**
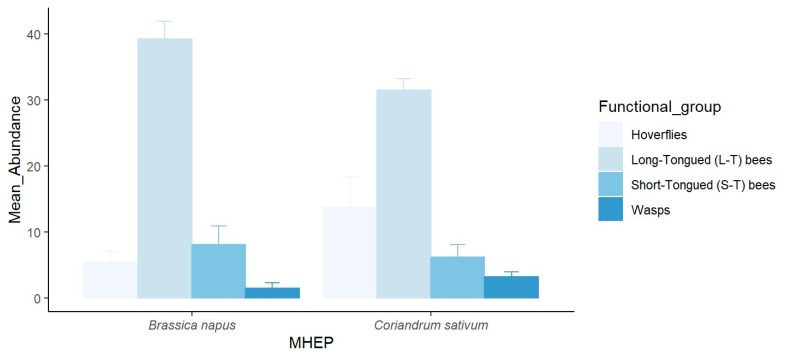
Mean abundance (±SE) of pollinator functional groups visiting each marketable habitat enhancement plant in large fields in 2021.

**Figure 3 insects-16-01164-f003:**
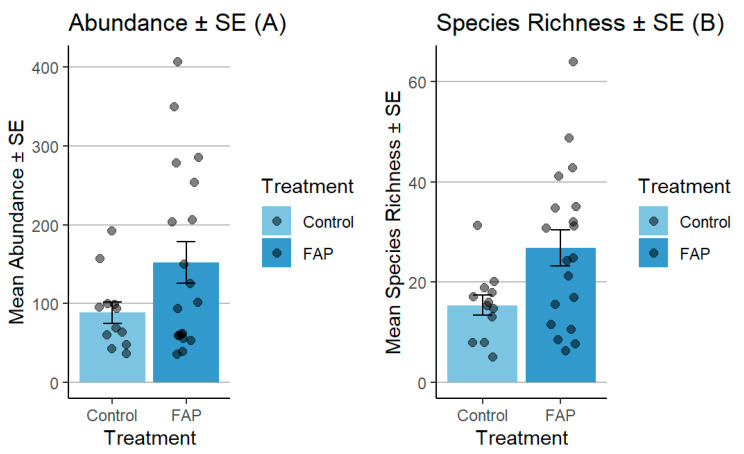
Comparison of pollinator abundance (**A**) and species richness (**B**) between FAP and control fields in small fields during 2018–2019. There were significant differences between FAP and control fields in terms of pollinator abundance (*p* < 0.05) and species richness (*p* < 0.01).

**Figure 4 insects-16-01164-f004:**
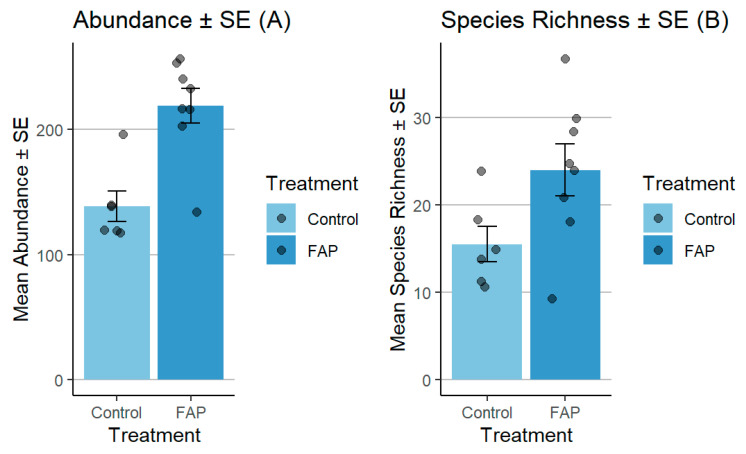
Comparison of pollinator abundance (**A**) and species richness (**B**) between FAP and control fields in large fields in 2021. There were significant differences between FAP and control fields in terms of flower visitor abundance (*p* < 0.001), and species richness (*p* < 0.05).

**Figure 5 insects-16-01164-f005:**
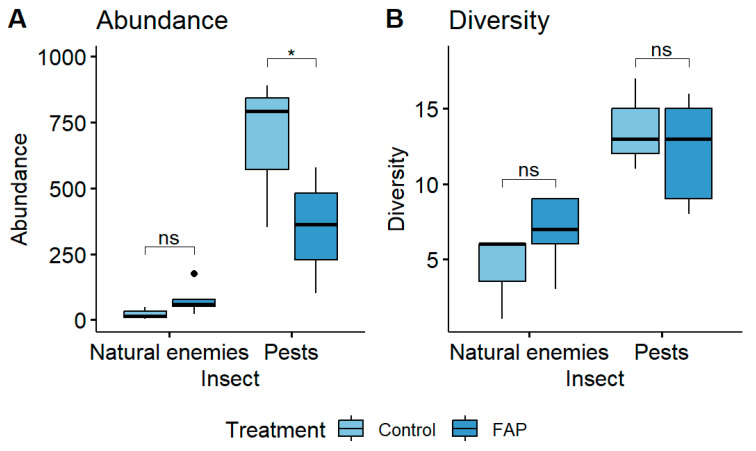
The abundance (**A**) and diversity (**B**) of natural enemies and pests of eight FAP (100% FAP field areas) and six control (100% control field areas) large fields. No significant differences in natural enemy abundance and diversity between FAP and control fields, but a significant difference in pest abundance in control fields (Emmeans). ns = not significant. Statistical analysis was performed by Emmeans tests, with asterisk indicating level of significance as follow: * *p* < 0.05.

**Figure 6 insects-16-01164-f006:**
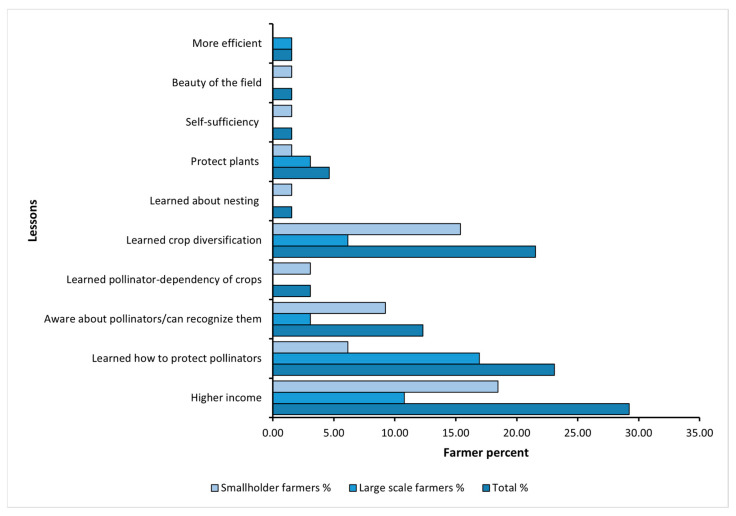
The lessons learned by the farmers from the trials implemented in small fields (2018–2019) and large fields (2021).

**Table 1 insects-16-01164-t001:** Marketable habitat enhancement plants used in sub-humid and semi-arid regions at different field scales, including their blooming duration.

Region	Family	Species	Blooming Period	Blooming Duration (Day)	Small Fields	Large Fields
Sub-humid	Fabaceae	*Vicia faba*	2 February to 1 April	58	-	2021
	Apiaceae	*Coriandrum sativum*	4 February to 1 April	56	-	2021
	Brassicaceae	*Brassica napus*	1 February to 5 April	62	-	2021
	Fabaceae	*Vicia faba*	31 January to 23 March	51	2018–2019	-
	Apiaceae	*Coriandrum sativum*	2 March to 2 April	31	2019	
	Brassicaceae	*Brassica napus*	10 February to 8 April	57	2018–2019	
	Lamiaceae	*Salvia hispanica*	5 February to 14 April	74	2018–2019	-
	Apiaceae	*Apium graveolens*	23 March to 14 May	53	2019	-
	Brassicaceae	*Eruca vesicaria*	8 February to 12 April	52	2018–2019	-
	Fabaceae	*Lupinus luteus*	26 February to 16 April	59	2018	-
	Fabaceae	*Lathyrus sativus*	6 March to 12 April	37	2019	-
	Fabaceae	*Medicago sativa*	2 March to 19 April	48	2018	-
	Fabaceae	*Lupinus albus*	24 February to 11 April	46	2018–2019	
Semi-arid	Fabaceae	*Vicia faba*	2 February to 1 April	58	-	2021
	Apiaceae	*Coriandrum sativum*	10 February to 1 April	48	-	2021
	Brassicaceae	*Brassica napus*	2 February to 5 April	60	-	2021
	Fabaceae	*Vicia faba*	1 February to 19 March	44	2018–2019	-
	Apiaceae	*Coriandrum sativum*	12 February to 21 March	35	2019	-
	Brassicaceae	*Brassica napus*	8 February to 1 April	50	2018–2019	-
	Fabaceae	*Lupinus albus*	13 February to 27 March	40	2018	-
	Fabaceae	*Medicago sativa*	23 March to 12 April	20	2018	-

**Table 2 insects-16-01164-t002:** The flower visitors of faba bean (*Vicia faba*) and marketable habitat enhancement plants (MEEP) from different semi-arid and sub-humid regions in small fields.

Region	Functional Group	Control	FAP	Region	Functional Group	Control	FAP
Semi-arid Settat	Long-tongue (L-T) bees	47	215	Sub-humid kenitra	Long-tongue (L-T) bees	519	1230
	Short-tongue (L-T) bees	225	723		Short-tongue (L-T) bees	58	331
	Syrphidae	21	102		Syrphidae	10	46
	Wasps	139	173		Wasps	16	70

**Table 3 insects-16-01164-t003:** The flower visitors of faba bean (*Vicia faba*) and marketable habitat enhancement plants (MHEP) from different semi-arid and sub-humid regions in large fields.

Region	Functional Group	Control	FAP	Region	Functional Group	Control	FAP
Semi-arid Settat	Long-tongue (L-T) bees	214	622	Sub-humid Sidi Slimane	Long-tongue (L-T) bees	392	473
	Short-tongue (L-T) bees	27	104		Short-tongue (L-T) bees	7	97
	Syrphidae	105	283		Syrphidae	33	66
	Wasps	28	54		Wasps	21	51

**Table 4 insects-16-01164-t004:** Economic assessment calculation of faba bean small fields in semi-arid Settat and sub-humid Kenitra regions in 2018 and 2019.

Year	Region	Treatment	Average Number of Harvested Faba Bean 75–Zone	Average Weight of Harvested Faba Bean in kg/75% Zone	Average Income from 75% Zone in MAD	Average Total Net Income from 25% Zone	Average Net Income from 0.03 ha in MAD
2018	Semi-arid Settat	FAP	8746	252	670	250	920
		Control	5091	153	318	20	338
		Diff%	72	64	110	1177	**172**
2019		FAP	6284	158	473	98	571
		Control	4225	92	275	34	309
		Diff%	49	72	72	187	**85**
2018	Sub-humid Kenitra	FAP	7898	304	1011	602	1612
		Control	8663	250	757	6	763
		Diff%	−9	21	34	10,035	**111**
2019		FAP	6493	254	762	195	957
		Control	5085	156	467	104	572
		Diff%	27.69	63	63	87	**67**

**Table 5 insects-16-01164-t005:** Economic assessment calculation of faba bean large fields in semi-arid Settat and sub-humid Sidi Slimane regions in 2021.

Region	Treatment	Average Number of Harvested Faba Bean 93–Zone	Average Weight of Harvested Faba Bean in kg/93% Zone	Average Income from 93% Zone in MAD	Average Total Net Income from 7% Zone	Average Net Income from 1 ha in MAD
Semi-arid Settat	FAP	292,206	6972	20,916	854	21,770
	Control	253,580	5387	16,162	1257	17,419
	Difference%	15	29	29	−32	**25**
Sub-humid Sidi Slimane	FAP	921,320	7651	19,127	965	20,092
	Control	694,090	5026	12,565	1100	13,666
	Difference%	33	52	52	−12	**47**

## Data Availability

The original contributions presented in this study are included in the article/Appendix A. Further inquiries can be directed to the corresponding author.

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
