# Peer review of "Insects’ and Farmers’ Responses to Pollinator-Related Habitat Improvement in Small and Large Faba Bean Fields in Morocco"

_insects, 2025, doi:10.3390/insects16111164_

Round 1
Reviewer 1 Report
Comments and Suggestions for Authors
In this study, the author evaluated the changes in Insects and Farmers’ income to pollinator-related habitat improvement in small and large Faba Bean Fields in Morocco. The study is significant to crop cultivation patterns in Morocco. However, the description of farmer’s net income provided in the result is overly simplistic, lacking sufficient details to fully illustrate Alternative Pollinators (FAP) approach is very economically important to the cultivation of faba bean in morocco. As mentioned by the authors in the Methods section, crop yield and market price were lacked in the Results section, which result in inability to understand the connection the diversity of pollinating insects and the yield of faba bean in this manuscript.
There are other issues:
- Figures should be labeled A and B,in figure 3 and 4; Figure 1 and 2 should be in result section
- Lind 85 and 87, two reference citation are same
- The pollinators should be classified in detail, and explained the impact of the increased diversity or amount of pollinators on faba bean in the FAP group.
- The references need to be checked, eg. Line622 nature-scientific reports
Author Response
Reviewer comment: In this study, the author evaluated the changes in Insects and Farmers’ income to pollinator-related habitat improvement in small and large Faba Bean Fields in Morocco. The study is significant to crop cultivation patterns in Morocco. However, the description of farmer’s net income provided in the result is overly simplistic, lacking sufficient details to fully illustrate Alternative Pollinators (FAP) approach is very economically important to the cultivation of faba bean in Morocco. As mentioned by the authors in the Methods section, crop yield and market price were lacked in the Results section, which result in inability to understand the connection the diversity of pollinating insects and the yield of faba bean in this manuscript.
Author’s response: Thank you for your positive feedback on the manuscript. We appreciate your acknowledgment of the extensive fieldwork conducted in this study. In the supplementary material, we put the tables that shows how we calculated the income. Table 1a and b are for the large fields, whereas table 1c,d,e and f were dedicated for the small fields, however we added a long paragraph please see lines 379-393.
There are other issues:
Reviewer comment: Figures should be labeled A and B in figure 3 and 4; Figure 1 and 2 should be in result section
Author’s response: Done, please see Figure 3 and 4. Concerning figure 1 and 2, we shifted them to result sections
Reviewer comment: Lind 85 and 87, two reference citation are same
Author’s response: Corrected, thank you for the remark
Reviewer comment: The pollinators should be classified in detail, and explained the impact of the increased diversity or amount of pollinators on faba bean in the FAP group.
Author’s response: All the specimens are well described in the supplementary material in Table 2, if we put them in the manuscript, it will be so long because, in total they are 6531 specimens
Reviewer comment: The references need to be checked, eg. Line622 nature-scientific reports
Author’s response: we checked the references once more based your remark, but the paper was published in Nature (scientific reports)
Reviewer 2 Report
Comments and Suggestions for Authors
This study analysed the impact of pollination service on income in small (300 m²) and large (1 ha) farms in Morocco.
The survey is a voice in the discussion about how future pollinator-protection programs for might look like. Interestingly, the authors compared the results of empirical experiments with the farmers' feelings.
In general, the study compared the diversity and richeness of pollinators and natural crop anemis in different fields located semi-arid and sub-humid zones. Different marketable habitat enhancement plants were used in the experiment.
Abstract is informative.
Introduction covers most immportant issues; aim of study is scratched, hypotheses presented. I found the experiment good desined. The description is quite easy to follow. Statistical analyses are described in details. Results are easy to follow.
Minor comments
Material and methods
L 118-119 Add the climate description according to international the Köppen classification.
L 141-144 – add the names of botanical families to each plant crop names .
Figures 1,2
Fonctionnel – is it correct – for me, not. It should be Functional group
Table 1 – Coriandrum sativum (small letter for sativum, please)
Author Response
Reviewer comment: This study assessed the impact of pollination service on income in small (300 m²) and large (1 ha) farms in Morocco.
The survey is a voice in the discussion about how future pollinator-protection programs might look like. Interestingly, the authors compared the results of empirical experiments with the farmers' feelings.
In general, the study compared the diversity and richness of pollinators and natural crop enemies in different fields located in semi-arid and sub-humid zones. Different marketable habitat enhancement plants were used in the experiment.
The abstract is informative.
Introduction covers most important issues; the aim of study is scratched, hypotheses presented. I found the experiment good designed. The description is quite easy to follow. Statistical analyses are described in detail. Finally the results are easy to follow.
Author’s response: Thank you very much for your positive and encouraging remarks about our work. We truly appreciate your feedback on entire manuscript, the clarity of the introduction, the design of the experiment, and the description of the analyses and results.
Minor comments
Material and methods
Reviewer comment: L 118-119 Add the climate description according to international the Köppen classification.
Author’s response: Added as you mentioned, please see line 119
Reviewer comment: L 141-144 – add the names of botanical families to each plant crop names.
Author’s response: We added botanical families, please see lines 143-148
Figures 1,2
Reviewer comment: Fonctionnel – is it correct – for me, not. It should be Functional group
Author’s response: Corrected please see the figures 1 and 2
Reviewer comment: Table 1 – Coriandrum sativum (small letter for sativum, please)
Author’s response: Corrected, thank you!
Reviewer 3 Report
Comments and Suggestions for Authors
I provide more specific for the authors to consider in chronological order.
Methods
Please add same climate precipitation details for the semi-arid region
In the analysis are you treating the 16 fields as independent even though they are in same farm among years? Should be a farm effect in models.
Line 139-145 - rearrange to have the presentation of the regions the same throughout from first mention (lines 116-120 through end) right now it flips back and forth.
Line 157- Just report what species they were replaced with.
Does this mean that the characteristics of the MHEPs were quite different between the years.
Line 160-162- Was each replicate field planted with both coriander and canola- this can be stated much more clearly.
Study is hard to interpret because the identities and diversity of the MHEPs different between field size and between years in small and large fields.
For the Large fields it is a bit confusingly worded how you report the sampling in the main crop versus the associated MHEP. Did the transects include both crop types cutting perpendicular across, or were they separate? More clarity is needed here.
Also consider adding the location of the transects to the supplemental figure,
See lines 215-216 for how this is done very clearly.
Lines 197-198. Grammar here is confused and makes it difficult to follow what you did.
Line 202 Delete preparation and just state directly all specimens were pinned and labeled
Why was the flower density not measured in the plantings. – what is the estimated difference in resources in a 1m2 area of canola or coriander versus fava bean? The discussion suggests differences in flowering duration and diversity. Could it also be that there simply were more flowers?
Table 1 Change to a small letter-s rather than a capital letter in Sativum
Analysis:
Year effect on small farms- Please clarify. Based on the methods above it seem like you resampled the same fam, but different fields at them in 2018 v 2019 (at least in some cases) this calls for some Farm ID random effect because the 2018-and 2019 data at those landscapes are not independent. You also have a potential year affect that should be included in a full model. More clarity is needed in the analysis.
Also need to provide the statistical model (write out model equations in the text).
Delete Lines 280-282?
Line 331 -338 I found the reporting of results confusing. The text reports summed total specimens with statistics and then figures shows mean per site. The stats are not performed on totals they are of differences per field with fields as replicates The intention of the design was to consider field independent then you should stick with the mean values alone rather than summed total. This is already done in early paragraphs. Essentially there are too many different summed values divided in different ways. The same issues is repeated in the lines 351- 361.
I would not make the layouts and styles of the figures different for Pollinators and pests. It isn’t clear what the point of doing this is and it adds to the confusion overall. Changing colors, styles, etc.
Lines 378 what is SM2?
In the farm interview results section, the results are a bit confusing because the same outcomes (or very similar ones) are listed twice with different importance (see lines 387- 388 and then 390), I believe this must be to do with them occurring in different parts of the interview (perhaps attributes, versus investment being two separate stages for questions?). I suggest greater detail and clarity in the methods in explaining the structure of the surveys.
Line 474 I suggest adding the Euro or other equivalent value difference not just the % in each case. It can be hard to track % increase versus % difference. For example 100% increase = doubling?
Line 425 -this statement is not accurate the increases are not really independent of field size nor was this directly tested in the study. I suggest a different wording here. I agree with the point that increases occur in larger field as well as in the smaller field sizes that have been more completely tested, so your general conclusion holds.
427- report what these are- do you mean field size and climate regions 2 X 2 = 4. Or is this from earlier work?
Line 431-436 is a very long and grammatically problematic sentence. Please break this up and present a bit more carefully.
Line 438 Apis mellifera should be italicized
Section 4.2 has quite a bit of redundancy in it. For example, you state that the crops are pollinator attractive a few times in a row, but there are also points that are restated from earlier paragraphs.
I was curious about the references for push-pull in line 465 and so looked them up and read as best I could. Ref 22 does not report a study of push-pull pest management, as far as I can tell neither do the other two. This raised concern for me as a reviewer. As I looked in other places, I found similar points where the citations seem to loosely match at best. Line 479 [citation 53] doesn’t really fit. As worded neither does [citation #57] in line 481. These last two could be used in a more general comparison, more like is done in the following sentence, Line 481-482. I suggest the authors very carefully go back through the document and consider the citations they are referencing and whether 1. they indeed are supportive of the point, 2. whether they make sense in the context of the sentence. Simply citing a paper that mentions the idea in passing at most is not really a valid citation. Additionally using a general citation when discussing specifics of your own system also would be mis-aligned.
Comments on the Quality of English Language
shown in the above comments
Author Response
I provide more specific for the authors to consider in chronological order.
Methods
Reviewer comment: Please add same climate precipitation details for the semi-arid region
Author’s response: We did the same wording, accordingly, thank you! Please see line 123.
Reviewer comment: In the analysis are you treating the 16 fields as independent even though they are in same farm among years? Should be a farm effect in models.
Author’s response: Thank you for your remark. However, this comment needs further clarification to be understood in context. If you are referring to the small fields, we had a total of 32 fields across both years (16 in Year 1 and 16 in Year 2), including 10 FAP and 6 control fields each year. The fields were located on different farms and were at least 1 km apart, but were conducted in the same manner, ensuring homogeneity and adequate distance between them. The results from the two years and two regions amount to 32 field (20 FAP fields and 12 control fields). We pooled the data from all fields separately (FAP and Control) since we considered them replicates. In our previous studies, we assessed the effects of region, climate, and landscape and found no significant differences between the two regions, indicating no landscape or climate effects on the community structure of pollinators and pests in our case. Furthermore, to ensure sufficient replicates for assessing the field-scale effect on biodiversity, we combined the data from the 20 FAP fields and 12 control fields.
Reviewer comment: Line 139-145 - rearrange to have the presentation of the regions the same throughout from first mention (lines 116-120 through end) right now it flips back and forth.
Author’s response: We did the same wording, accordingly, thank you!
Reviewer comment: Line 157- Just report what species they were replaced with.
Does this mean that the characteristics of the MHEPs were quite different between the years.
Author’s response: We added the MHEPs replaced with accordingly, yes some of them are changed in the second year in small fields based on the recommendations of the farmers. Please see line 162.
Reviewer comment: Line 160-162- Was each replicate field planted with both coriander and canola- this can be stated much more clearly.
Author’s response: Yes, the ensemble of FAP fields in large fields were planted with coriander and canola. We changed some wording to make it clearer.
Reviewer comment: Study is hard to interpret because the identities and diversity of the MHEPs different between field size and between years in small and large fields.
Author’s response: MHEP play a crucial role in the field component of FAP, and their diversification adds significant value to the approach. Here, we used two different field sizes, but we always compared FAP and control in small and large fields separately.
Reviewer comment: For the Large fields it is a bit confusingly worded how you report the sampling in the main crop versus the associated MHEP. Did the transects include both crop types cutting perpendicular across, or were they separate? More clarity is needed here.
Also consider adding the location of the transects to the supplemental figure,
See lines 215-216 for how this is done very clearly.
Author’s response: we added the locations of the nets in the experimental designs in excel file in the main crop and in MHEP. Regarding the sampling in the main crop, we did two distanced transects, one in the center and one at the edge to have an idea about the entire field.
Reviewer comment: Lines 197-198. Grammar here is confused and makes it difficult to follow what you did.
Author’s response: thank you for the remark, please see lines 203-204 it is clearer than before.
Reviewer comment: Line 202 Delete preparation and just state directly all specimens were pinned and labeled
Author’s response: Changed accordingly, please see line 207
Reviewer comment: Why was the flower density not measured in the plantings. – what is the estimated difference in resources in a 1m2 area of canola or coriander versus fava bean? The discussion suggests differences in flowering duration and diversity. Could it also be that there simply were more flowers?
Author’s response: The FAP approach is a farmer-centered approach; we used the same methods as the farmers to make it easier for them to do it independently in the future. We did not use quadrats (1x1m2) in 7% of the FAP (MHEP) or control (faba bean) fields. Instead, we calculated the total area directly. We believe this point was unclear before, but we have now added a paragraph in the Materials and Methods section explaining how the income was calculated
Reviewer comment: Table 1 Change to a small letter-s rather than a capital letter in Sativum
Author’s response: Thank you for the remark. We changed it accordingly.
Reviewer comment: Analysis:
Year effect on small farms- Please clarify. Based on the methods above it seem like you resampled the same fam, but different fields at them in 2018 v 2019 (at least in some cases) this calls for some Farm ID random effect because the 2018-and 2019 data at those landscapes are not independent. You also have a potential year affect that should be included in a full model. More clarity is needed in the analysis.
Also need to provide the statistical model (write out model equations in the text).
Author’s response: Thank you for this valuable comment. We would like to clarify that although we sampled different fields within the same area, the landscapes and soil characteristics were highly homogeneous, which minimized environmental variability between sites. For this reason, we pooled the data from all FAP fields and all control fields to calculate abundance and species richness. Regarding the year effect, we acknowledge that data were collected in both 2018 and 2019, which introduces potential non-independence of observations across years. However, the year was not among the objectives of comparison, as all fields were managed using the same standardized sampling methods. Therefore, we did not include the year as an effect in our analysis. Our main objective was to assess the differences between FAP and control fields, without including the year effect. We appreciate your suggestion and will consider it for future manuscripts with objectives focusing on temporal variation.
Reviewer comment: Delete Lines 280-282?
Author’s response: Thank you for the remark.
Reviewer comment: Line 331 -338 I found the reporting of results confusing. The text reports summed total specimens with statistics and then figures shows mean per site. The stats are not performed on totals they are of differences per field with fields as replicates The intention of the design was to consider field independent then you should stick with the mean values alone rather than summed total. This is already done in early paragraphs. Essentially there are too many different summed values divided in different ways. The same issues is repeated in the lines 351- 361.
Author’s response: Yes, you are right. The text reports the total number of specimens in FAP and control fields, as well as the species richness found in each field. Therefore, both figures present the same information, showing the total number of specimens and the diversity between FAP and control fields
Reviewer comment: I would not make the layouts and styles of the figures different for Pollinators and pests. It isn’t clear what the point of doing this is and it adds to the confusion overall. Changing colors, styles, etc.
Author’s response: we changed color and the style accordingly and do the same as pollinators like pests and natural enemies
Reviewer comment: Lines 378 what is SM2?
Author’s response: It is supplementary material 2, it is corrected now.
Reviewer comment: In the farm interview results section, the results are a bit confusing because the same outcomes (or very similar ones) are listed twice with different importance (see lines 387- 388 and then 390), I believe this must be to do with them occurring in different parts of the interview (perhaps attributes, versus investment being two separate stages for questions?). I suggest greater detail and clarity in the methods in explaining the structure of the surveys.
Author’s response: Yes, you are right in that part the question was similar, we deleted the second phrase. However, if we go further and mentioned all the questions details, the reader will be lost because they are a lot of them. We summarize only the important ones and if the reader is interested in the survey the supplementary material retains the entire questions of the survey.
Reviewer comment: Line 474 I suggest adding the Euro or other equivalent value difference not just the % in each case. It can be hard to track % increase versus % difference. For example 100% increase = doubling?
Author’s response: We understand your point, however here we are comparing both two field types (FAP vs. control), we found that FAP fields increased 36% compared to control fields in large fields, and we did the same for small fields. So, we think the percentage is much clearer than the Euro or MAD (local currency). Besides, the tables 1a,b,c,d,e,f are showing all the details even about the price.
Reviewer comment: Line 425 -this statement is not accurate the increases are not really independent of field size nor was this directly tested in the study. I suggest a different wording here. I agree with the point that increases occur in larger field as well as in the smaller field sizes that have been more completely tested, so your general conclusion holds.
Author’s response: We changed accordingly, please see lines 462-463
Reviewer comment: 427- report what these are- do you mean field size and climate regions 2 X 2 = 4. Or is this from earlier work?
Author’s response: Right, it is from earlier work on the same approach, thanks for the remark. It is mentioned in the text
Reviewer comment: Line 431-436 is a very long and grammatically problematic sentence. Please break this up and present a bit more carefully.
Author’s response: It is done.
Reviewer comment: Line 438 Apis mellifera should be italicized
Author’s response: Done, thank you!
Reviewer comment: Section 4.2 has quite a bit of redundancy in it. For example, you state that the crops are pollinator attractive a few times in a row, but there are also points that are restated from earlier paragraphs.
Author’s response: We have removed redundant parts to make the section more concise and effective. Please see the revised section. Thank you!
Reviewer comment: I was curious about the references for push-pull in line 465 and so looked them up and read as best I could. Ref 22 does not report a study of push-pull pest management, as far as I can tell neither do the other two. This raised concern for me as a reviewer. As I looked in other places, I found similar points where the citations seem to loosely match at best. Line 479 [citation 53] doesn’t really fit. As worded neither does [citation #57] in line 481. These last two could be used in a more general comparison, more like is done in the following sentence, Line 481-482. I suggest the authors very carefully go back through the document and consider the citations they are referencing and whether 1. they indeed are supportive of the point, 2. whether they make sense in the context of the sentence. Simply citing a paper that mentions the idea in passing at most is not really a valid citation. Additionally using a general citation when discussing specifics of your own system also would be mis-aligned.
Author’s response: We would like to thank you for this precise comment. I believe there was a mistake, but it is caused by Mendeley. The issue with line 465 (Ref 22) has now been deleted. However, the other references indeed refer to the push-pull approach, as I am one of the authors of those papers. We have rechecked all the references, and they are now correct.
Reviewer 4 Report
Comments and Suggestions for Authors
Dear colleagues!
The presented work is original. The problem of biodiversity conservation on the planet is one of the global problems of mankind. The presented work reveals not only the fundamental aspect of the problem, but also provides relevant information on the applied part, even indicating the economic benefits.The authors have done a great job. I was surprised by the scale of the research, the well-chosen techniques, the use of a variety of entomophilic plants in experiments, and the huge amount of entomological material. To process the data, the authors applied a variety of statistical methods using a modern statistical data analysis package.Undoubtedly, the work should be published. It is difficult for me to make critical comments to the authors of the manuscript. I recommend that the authors apply at least one statistical method for analyzing the results of the farmer survey.
Author Response
Reviewer comment: Dear colleagues!
The presented work is original. The problem of biodiversity conservation on the planet is one of the global problems of mankind. The presented work reveals not only the fundamental aspect of the problem, but also provides relevant information on the applied part, even indicating the economic benefits.The authors have done a great job. I was surprised by the scale of the research, the well-chosen techniques, the use of a variety of entomophilic plants in experiments, and the huge amount of entomological material. To process the data, the authors applied a variety of statistical methods using a modern statistical data analysis package.Undoubtedly, the work should be published. It is difficult for me to make critical comments to the authors of the manuscript. I recommend that the authors apply at least one statistical method for analyzing the results of the farmer survey.
Author’s response: We sincerely thank you for your positive and encouraging feedback on our manuscript. We greatly appreciate your recognition of the extensive fieldwork and the methodological approaches used in this study. Your constructive suggestion regarding the application of a statistical method for analyzing the farmer survey results is also highly valued, and we carefully consider it in order to further strengthen the manuscript. Please see the statistical analysis section in the manuscript
Round 2
Reviewer 1 Report
Comments and Suggestions for Authors
The manuscript has been revised and improved, but further modifications are still required.
Yield data were collected from 2018 or 2019, whereas all values are priced at 2021 levels; consequently, yield can serve as a reliable indicator of net income. Moreover, this should be the primary positive response to pollinator-related habitat improvement in small and large faba-bean fields reported by farmers. Therefore, the authors should include the yield data in the manuscript.
The author mentioned”All the specimens are well described in the supplementary material in Table 2, if we put them in the manuscript, it will be so long because, in total they are 6531 specimens”, however the authors did not describe the details of species difference in manuscript, like figure 3 just mentioned the total species numbers. Because, certain insect species might be important for the faba bean yield; that is important response from insects.
The journal name should be “Scientific reports” instead of nature-scientific reports
Line 393 “m2” , 2 should be superscript
Author Response
Reviewer comment: The manuscript has been revised and improved, but further modifications are still required.
Yield data were collected from 2018 or 2019, whereas all values are priced at 2021 levels; consequently, yield can serve as a reliable indicator of net income. Moreover, this should be the primary positive response to pollinator-related habitat improvement in small and large faba-bean fields reported by farmers. Therefore, the authors should include the yield data in the manuscript.
Author’s response: We have included the yield data and divided the tables for small and large fields to make them clearer. However, we did not include the description, as it is the same as for the large fields, to avoid redundancy. Thank you!
Reviewer comment: The author mentioned”All the specimens are well described in the supplementary material in Table 2, if we put them in the manuscript, it will be so long because, in total they are 6531 specimens”, however the authors did not describe the details of species difference in manuscript, like figure 3 just mentioned the total species numbers. Because, certain insect species might be important for the faba bean yield; that is important response from insects.
Author’s response: We have now included all specimens collected from both FAP and control fields, as well as from the different regions, as advised. Thank you!
Reviewer comment: The journal name should be “Scientific reports” instead of nature-scientific reports
Author’s response: Edited as advised, thank you!
Reviewer comment: Line 393 “m2” , 2 should be superscript
Author’s response: Edited as advised, thank you!